# Video-Based Parking Occupancy Detection for Smart Control System

**Lun-Chi Chen [1] , Ruey-Kai Sheu [2] , Wen-Yi Peng [2], Jyh-Horng Wu [3] and Chien-Hao Tseng [3],***

[1]   College of Engineering, Tunghai University, Taichung 40704, Taiwan; lunchi@thu.edu.tw
[2]   Department of Computer Science, Tunghai University, Taichung 40704, Taiwan;
    rickysheu@thu.edu.tw (R.-K.S.); s05352017@go.thu.edu.tw (W.-Y.P.)
[3]   National Center for High-Performance Computing, National Applied Research Laboratories, Hsinchu 30076,
    Taiwan; jhwu@nchc.narl.org.tw
*   Correspondence: 0903049@nchc.narl.org.tw; Tel.: +886-4-2462-0202 (ext. 855)

**Abstract:** Street lighting is a fundamental aspect of security systems in homes, industrial facilities, and public places. To detect parking lot occupancy in outdoor environments, street light control plays a crucial role in smart surveillance applications that can perform robustly in extreme surveillance environments. However, traditional parking occupancy systems are mostly implemented for outdoor environments using costly sensor-based techniques. This study uses the Jetson TX2 to develop a method that can accurately identify street parking occupancy and control streetlights to assist occupancy detection, thereby reducing costs, and can adapt to various weather conditions. The proposed method adopts You Only Look Once version 3 (YOLO v3, Seattle, WA, USA) based on MobileNet version 2 (MobileNet v2, Salt Lake City, UT, USA), which is area-based and uses voting to stably recognize occupancy status. This solution was verified using the CNRPark + EXT dataset, a simulated model, and real scenes photographed with a camera. Our experiments revealed that the proposed framework can achieve stable parking occupancy detection in streets.

**Keywords:** parking occupancy detection; You Only Look Once; Jetson TX2; smart streetlight; control system

---

## 1. Introduction

With the development of technology, infrastructure has been gradually improving; city infrastructure is vital for ensuring social, economic, and physical welfare. Hence, cities face considerable urban planning challenges [1]. They face rapidly growing populations as well as social and sustainability changes. Many approaches to automating and facilitating smart cities have been developed. Smart cities are typically described as complex networks formed from resource interdependencies. Recent advancements in computer vision and intelligent technologies may assist urban renewal and reduce social costs in the pursuit of improving welfare [2–4].

To improve living environments using an existing device and vision-based technology, we propose a method of vehicle occupancy detection using a streetlight assistant system. The system detects vehicles with streetlight cameras and finds empty parking spaces on roadsides to reduce the amount of time required to find available parking spaces. It uses streetlights to assist detection by automatically maintaining half brightness at night and shifting to high brightness when an object is detected. Stability is a primary factor that should be considered when selecting a detection scheme. Stable indicators should accurately provide information about parking spaces on streets, regardless of variables in the environment, such as rain, difficult angles of view, and nighttime conditions. Second, maintenance is a crucial determinant, which includes the difficulties of deploying and maintaining new tools for every

parking space. Third, the number of sensing units directly affects the establishment and maintenance expenses of detection systems.

Traditional lighting systems only offer two options, i.e., on and off, which wastes energy [5]. Hence, more innovative energy-saving devices have been developed. In this regard, several automatic smart streetlight methods have been proposed for control at night using light-dependent resistor (LDR) sensors [6] to detect vehicle and human movement, switching on streetlights ahead of the object and switching them off subsequently to save energy. Common visual sensors such as infrared and obstacle avoidance sensors [7] have also been implemented. Sun tracking sensors [8] are also used to switch off streetlights by detecting sunlight luminance. Jagadeesh et al. [9] presented a method using LDR sensors and image processing to detect vehicle movement. The method [10] proposed a solar energy-based and ZigBee-based system. Mumtaz et al. [11] introduced an Arduino-based automation system, which used solar rays and object detection to control streetlights through LDR and infrared sensors. In addition to on and off settings, it introduced a half brightness (dim) setting [12,13] to control streetlight intensity by dimming or brightening the light intensity according to the detection results.

Many parking guidance systems are currently on the market, but traffic congestion remains persistent everywhere. Maintaining smooth traffic conditions concerns city managers. In the parking detection literature, researchers have proposed methods that use sensors, traditional image processing, and deep learning. Sensor-based methods, such as ultrasonic sensing, inductive looping, infrared laser sensing, and magnetic signaling, are preinstalled in parking grids [14,15] to determine occupancy status. Marso et al. [16] proposed a parking-space occupancy detection system using Arduino, Raspberry Pi, and combined infrared sensors with ultrasonic sensors for exterior/interior usage to detect whether a parking space was occupied. These sensors are typically reliable; however, a disadvantage of sensor-based methods is the considerable cost of applying the technology to entire streets. Furthermore, sensor detection of parking spaces is typically applied in indoor parking lots. Each grid requires at least one sensor and needs a server device to compute the parking gird. Thus, the implementation of this on streets is unfeasible. The expense of establishing and maintaining such a detection system may directly affect its quality, and installing hundreds of sensors at large or outdoor parking areas is impracticable. Computer vision-based methods reduce costs and display real-time sequence images to observe multiple objects. Studies [17,18] have proposed image-based vehicle detection methods, image fragmentation into gray levels, and segment area distribution analysis to detect the presence of vehicles. However, traditional image processing has difficulties with processing complex backgrounds such as streets, bright or dark targets, and large areas with occlusion, as well as with separating targets and backgrounds.

Several machine learning algorithms and object classification methods have been developed over many years. One system [19] uses small camera modules based on Raspberry Pi Zero and computationally-efficient algorithms for occupancy detection based on a histogram of oriented gradient feature descriptors and support vector machine classifiers. Advancements in deep learning have revolutionized the machine learning field, enabling unparalleled performance and automatic feature extraction from images without artificial intervention, in contrast to most traditional machine learning algorithms. For parking detection, Acharya and Khoshalham [20] proposed a system for real-life situations by providing functionalities for real-time monitoring, and by modifying a convolutional neural network (CNN) to classify whether the space is occupied or free. Nurullayev and Lee [21] proposed methods based on a CNN designed to detect parking grid occupancy in parking lots by using an open dataset to evaluate a detection model. Object detection can produce the same effects, and many methods apply object detection in parking sections, such as Faster region CNN (R-CNN) [22], single-shot detection [23], and You Only Look Once (YOLO) [24]. Solutions based on Mask R-CNN [25,26] have been proposed for a space-based method that relies on classifying parking space occupancy. This method requires the hand-labeling of a specific parking scene and the training of a model that may not be generalizable to other parking scenes. A traditional object detection method

with a Haar cascade-based [27] approach was used to detect parking spaces, and was implemented with CNN.

Methods using deep network methods focus on object classification and identifying dataset labels, such as classifying parking space occupation. However, they incompletely detect entire scenes. Our goal was to detect all objects at once instead of a single parking grid. Thus, to build a robust parking occupancy detection system that implements a YOLO v3 [28] detection model with MobileNet v2 [29], many difficulties must be considered, such as interobject occlusion, unfavorable weather, and camera angle of views that cause judgment errors. Hence, adapting to specific situations in every street parking space is difficult.

To evaluate and compare methods in real-world situations, we trained and tested them on the publicly available dataset, CNRPark + EXT [30] (the data viewpoint like Figure 1). In the dataset, label data from the CNN model and parking grid coordinates and image data are provided. In this study, we used LabelImg [31] to create training data for object detection with the YOLO model. In general, parking space detection may encounter different environmental constraints, as shown in Figure 2. The proposed method presented more reliability than did prior works, particularly when tested on a completely different subset of images. In a study, method performance was compared with the well-known architecture, AlexNet [32]. Our model also demonstrated highly promising performance compared with AlexNet. Moreover, our investigations indicated that our approach was more robust, stable, and well-generalized for unseen images captured from different camera viewpoints than were previous approaches for classifying parking space occupancy. Our approach displayed strong indications that it could be effectively generalized to other parking lots.

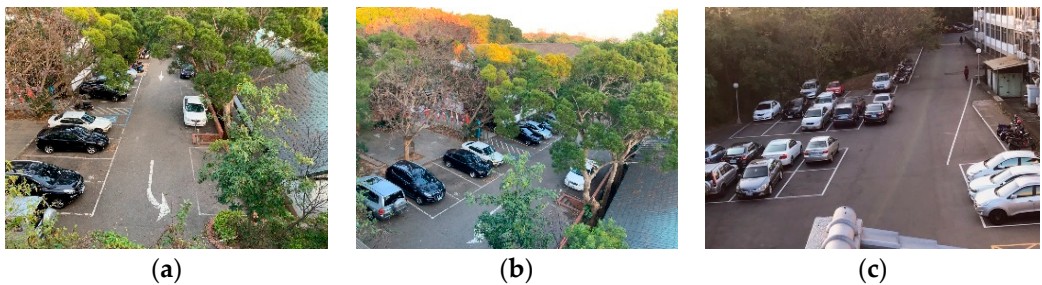

(**a**)          (**b**)          (**c**)

**Figure 1.** The real-world presented large variations in appearance, occlusions, and displayed different camera angle: (**a**) Horizontal view, (**b**) Side view, and (**c**) Vertical view.

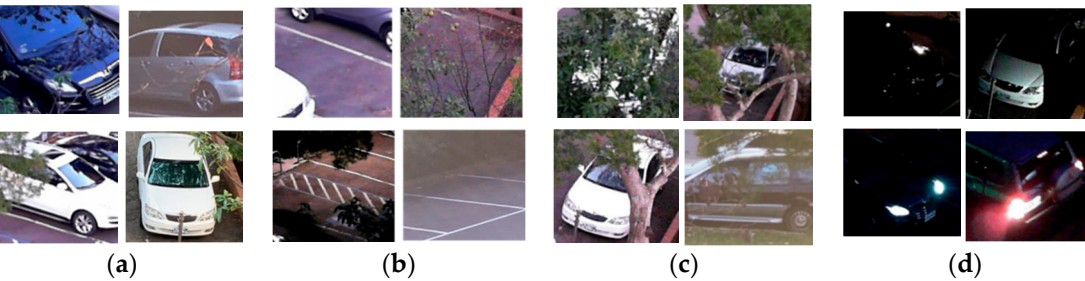

(**a**)          (**b**)          (**c**)          (**d**)

**Figure 2.** Actual parking spaces in label patches. In the real-world four categories may be encountered: (**a**) Occupied, (**b**) Vacancy, (**c**) Occlusion, and (**d**) dark or nighttime.

In this paper, a method for efficient street-side parking occupancy detection on embedded devices is proposed, which is easily deployed on any street using existing roadside equipment and real-time processing on a Jetson TX2 [33–35]. The method uses existing surveillance cameras and embedded devices to produce an efficient, high-speed, and lightweight detection model. To find free parking spaces, the model supports one-shot object detection to find vehicles, and uses image processing to identify parking grid occupation from a sequence image. At night, streetlights assist in detecting

the object, provide on-demand adaptive lighting, and make streetlights adjust their brightness based on the presence of pedestrians, cyclists, or cars. Furthermore, the proposed system detects vehicle occupancy with deep learning that uses various operations to check each parking space. This method, however, may represent a burden to the hardware. Therefore, our novel approach was proposed for existing streetlight cameras that use technology with deep learning and match with local computers to calculate occupancy results for parking spaces on entire roadsides.

This system compares with actual sensor system approaches. The sensor may accurately sense every parking space in real-time, but each parking grid requires at least one sensor device. This would increase the difficulty of construction and maintenance and even cause expensive costs. Our method uses existing streetlights with embedded systems to build a visual-based model. The model intelligently detects all objects at once, instead of each parking space individually, and adapts to a wide range of light conditions and angles of view. Due to the fact that visual-based methods are easily affected by the environment, this paper proposes an algorithm to detect parking occupancy from a wide range of perspectives and employs existing streetlights.

The rest of this paper is structured as follows: Section 2 explains the method in detail and provides the experimental results of a laboratory-scale prototype. Section 3 describes the experiments and results. Finally, Section 4 presents the conclusions.

## 2. Materials and Methods

This section describes our parking detection system on street scenes. This system includes materials, methods, vehicle detection, occupancy identification with the voting mechanism in postprocessing, and streetlight control at night to assist in detection, as shown in Figure 3. A diagram for the proposed system using an existing camera-based Jetson TX2 is depicted in Figure 4. The architecture diagram shows that we received real-time images from a camera mounted on a streetlight. All operations, including detection, identification, and control, are processed on the Jetson TX2.

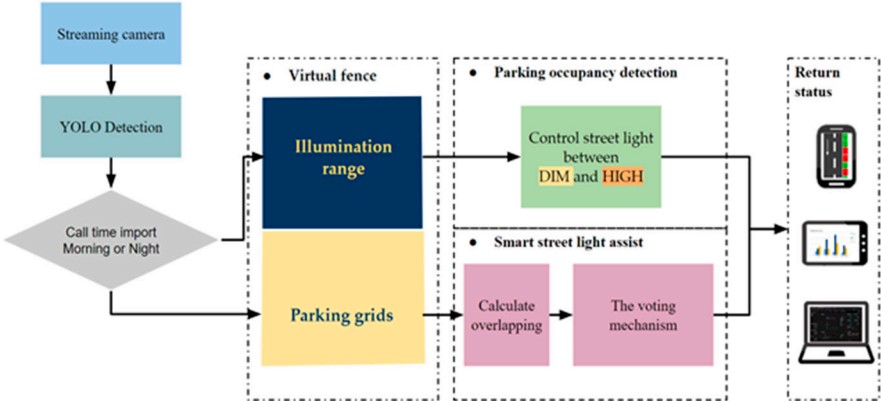

**Figure 3.** System process chart.

The proposed methodology is shown in Figure 5. The input video uses embedded hardware to directly connect to existing street surveillance. Human and vehicle detection involves high-performance deep learning algorithms, while the YOLO of the bounding box and parking grid calculate overlapping areas according to the intersection-area-to-voting-mechanism ratio from the sequence image to identify the occupancy status. The streetlights automatically light up from DIM settings from the detection results. All steps are processed by the Jetson TX2.

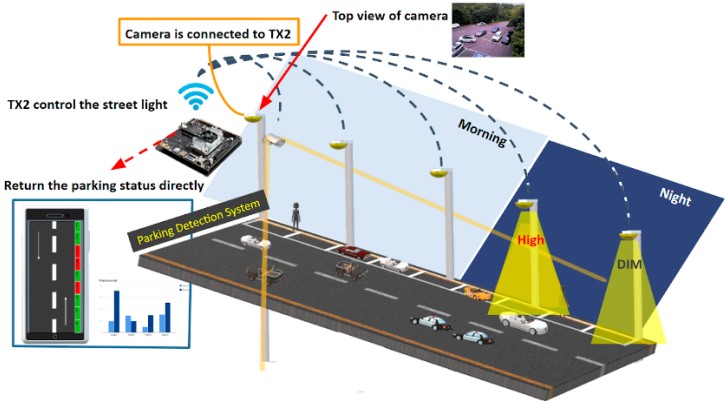

**Figure 4.** Architecture design of the automatic streetlight and smart street parking control system.

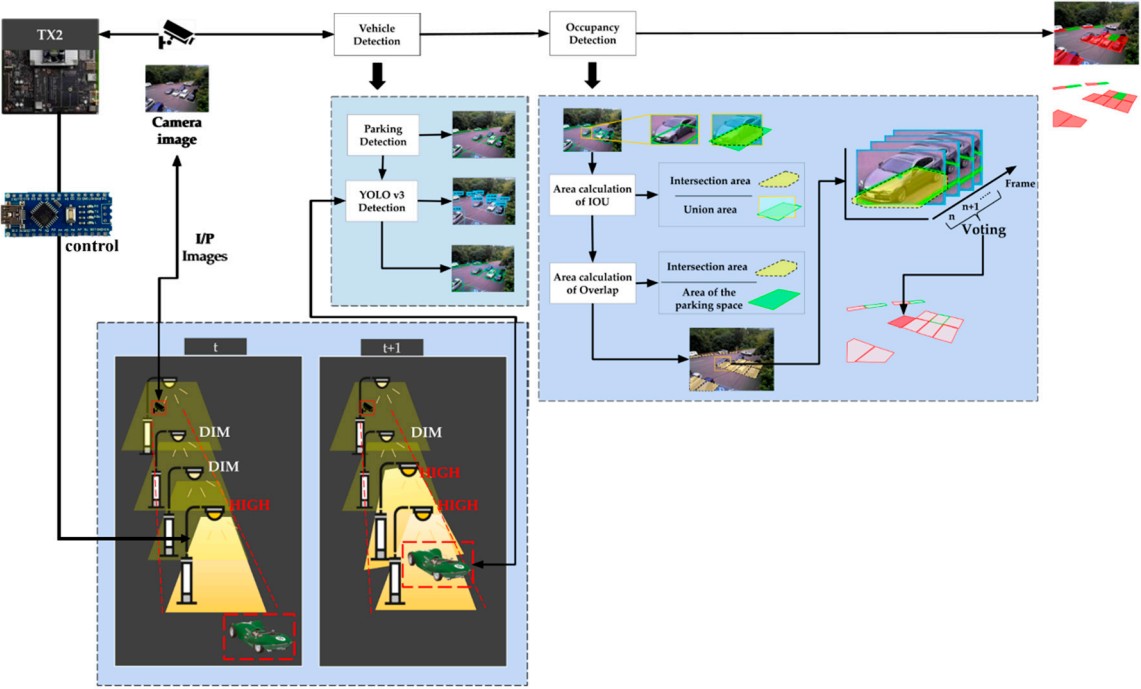

**Figure 5.** System methodology.

## 2.1. Jetson TX2

Jetson TX2 is NVIDIA's Compute Unified Device Architecture (CUDA)-capable embedded artificial intelligence (AI) computing device, which has high power efficiency. This supercomputing module brings genuine AI processing to end devices, and has low power consumption. It is equipped with a graphics processing unit (GPU)-based board with NVIDIA 256 core pascal architecture and a 64-bit hex core ARM (version 8) central processing unit (CPU), which contains a specialized architecture for AI applications. It enables real-time deep learning applications in small form-factor products such as drones. Jetson TX2 has 8 GB of low-power double data rate synchronous dynamic random access memory and supports PCIe (version 2.0) and various peripherals such as general-purpose inputs/outputs (GPIOs), display, camera, universal serial buses (version 3.0 and 2.0), Ethernet, and 802.11ac wireless local area networks. Therefore, creating intelligent applications quickly and anywhere for prototyping is simple. NVIDIA provides the required drivers and CUDA toolkits for Jetson TX2 through the JetPack software development kit (version 4.2), which is used to automate basic installations for Jetson TX2, and includes Board Support Packages and libraries specialized in

deep learning and computer vision. These features of the GPU-based board helped us implement the parking detection system in street scenes and in the implementation on the streetlight.

The high-performance, low-energy computing capabilities of deep learning and computer vision provided an ideal platform for embedded projects with high computing demands. To use the deep learning method to support the entire architecture, we moved object detection, image judgment, and analysis onto the Jetson TX2 to achieve parking space detection and streetlight control. The image-based system was a valid choice.

To directly receive real-time images interfacing the street camera, these data aid in tracking the statuses of the streetlight and parking grids from the roadside. The Jetson TX2, which receives live feed images, compares images at the time and recalls the deep learning model to detect moving objects. To simulate the environment of the detection model and street lamp, we used a webcam to receive real-time images and use the GPIO pin of the Jetson TX2 to control six light-emitting diode (LED) lights (see Figure 6).

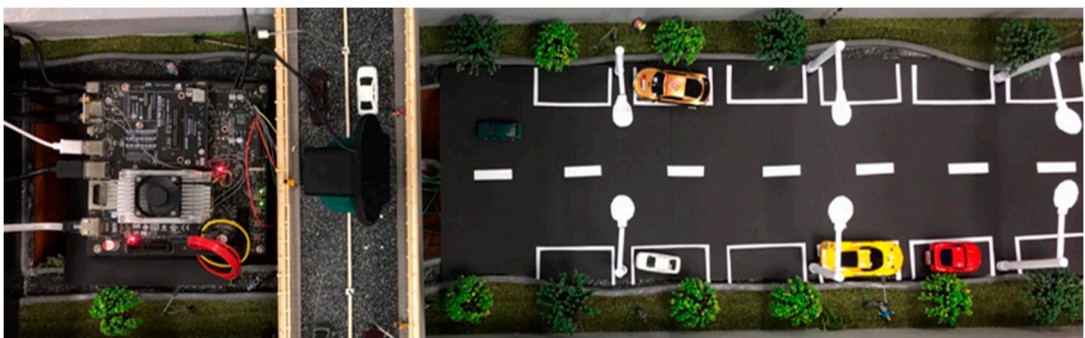

**Figure 6.** Streetlight model.

## 2.2. Detection Model

### 2.2.1. YOLO

Traditional object detection uses a sliding window to detect whether the subwindow has detected a target. However, methods that rely on classifiers and sliding windows to detecting targets may require long-term training and produce too many bounding boxes. The purpose of a single-shot object recognition algorithm such as YOLO is to achieve immediate recognition without sacrificing accuracy. These algorithms are able to handle complex tasks, such as pedestrian, license-plate, vehicle, and traffic-sign detection [36–41]. YOLO is an end-to-end object detection network. Unlike R-CNNs [42], the proposed technique does not rely on regional proposals. Instead, it detects objects by using CNN and cuts the original image into an $S \times S$ nonrepetitive network. A grid image in which $S$ varies according to image size and various anchor boxes is generated in each grid, and the neuron learns the offset of anchor boxes from the bounding boxes of actual objects. This method can detect bounding boxes and categories of multiple objects within images without region proposals or resorting to simplistic regression of entire images to bounding boxes.

$$IOU = \frac{DectionResult \cap GroundTruth}{DectionResult \cup GroundTruth} \tag{1}$$

YOLO cuts the original image into $S \times S$ nonrepetitive grid images, where $S$ is the number of grids determined by users. The neural network uses anchor boxes to estimate the center of vehicles in a grid that is responsible for predicting vehicles. In each grid, the bounding box is estimated to generate four coordinates: $t_x, t_y, t_w,$ and $t_h$; $t_x$ and $t_y$ represent the central coordinates of the predicted bounding boxes. $t_w$ and $t_h$ represent the width and height of the predicted bounding boxes, respectively.

YOLO's actual operation is as follows: For each grid, the model predicts five variables, i.e., $t_x, t_y, t_w, t_h$, and $t_o$, and $t_x$ and $t_y$ for ground truth and anchor center point offsets $t_w$ and $t_h$ for length and width, respectively, to represent object confidence. As displayed in Figure 7, $t_x, t_y, t_w, t_h$, and $t_o$, and $t_x$ and $t_y$ for ground truth and anchor center point offsets $t_w$ and $t_h$ for length and width, respectively, to represent object confidence. As displayed in Figure 7, $b_x, b_y, b_w$, and $b_n$ are the actual bounding box of the objects. When cells deviate from the upper left of insulator images by $(c_x, c_y)$ then $p_w$ and $p_h$ are the length and width of the anchors, respectively. The bounding box prediction standard is intersection over union (IOU), which is used to measure the accuracy of object detectors on datasets. The optimal situation is IOU = 1, signifying that a predicted bounding box completely coincides with the bounding box of a real insulator.

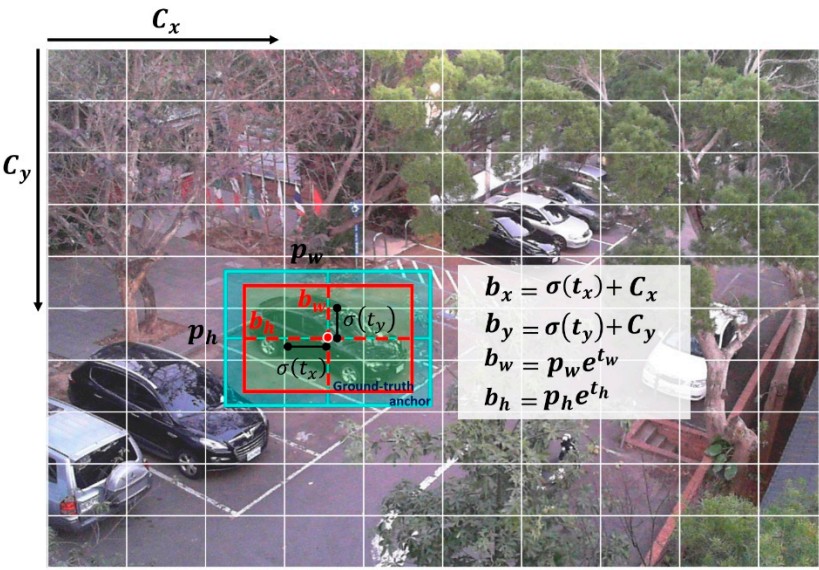

**Figure 7.** Relationship between anchor box and ground truth.

The characteristics of the overlapping area are used in bounding boxes, which fits real objects to calculate the area. To enhance the detection accuracy of small objects, YOLO v3 uses a Feature Pyramid Network [43] upsampling and fusion approach for object detection, signifying top-down architecture with lateral connections developed for building high-level semantic feature maps on all scales. YOLO is based on Darknet-53 [28], adopts full convolution, and takes advantage of the residual structure of the residual neural network (ResNet) [44] to enable model fitting on the embedded device and lighter and accurate detection with YOLO. Darknet-53 is trained on ImageNet for 1000 object types [45]. In deep learning, the general trend is to increase network depth and computation complexity to increase the network performance. However, several real-time applications, such as self-driving cars, augmented reality, and real-time object recognition, require faster networks. To run YOLO on the Jetson TX2, we simplified the network architecture of YOLO v3 and increased the computing speed without significantly reducing the accuracy. We used MobileNet [46] to optimize the YOLO model.

2.2.2. MobileNet

Deep learning requiring high-performance computers to achieve fast computing is a common bottleneck because the depth of neural networks requires many parameters and is poorly-suited for use on embedded devices. MobileNet reduces the computational complexity of CNNs, allowing deep learning to be performed under limited hardware conditions to achieve the desired effect. The basic convolution structure of MobileNet is presented in Figure 8. Figure 8a reveals the conventional convolutional filter, which is four-dimensional. MobileNet proposes a separable convolution structure that replaces the original convolutional layer. It consists of a depthwise convolutional layer and

pointwise convolution layer. Figure 8b displays the depthwise convolutional layers using $3 \times 3$ kernels, with each kernel only iterating one channel of an image, unlike traditional convolution, in which a set of kernels iterates all channels of an image. Figure 8b displays the common pointwise convolutional layers by using $1 \times 1$ kernels. These layers have smaller computational costs than do standard convolution layers. The depthwise separable convolution layers function on the factorization principle.

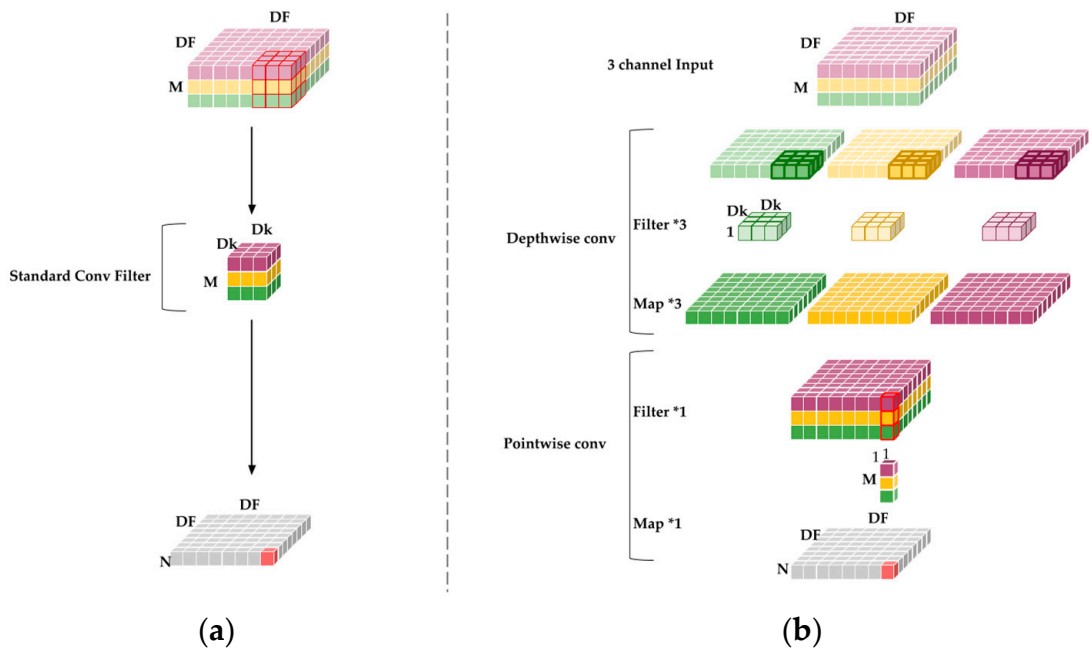

**Figure 8.** MobileNet architecture diagram. (**a**) Standard convolution with $3 \times 3$ kernel. (**b**) Convolution operation is replaced with depthwise and pointwise convolution.

The first layer of the architecture is fully convoluted; in addition, the remainder of the MobileNet structure is built on depthwise separable convolutions. The final fully connected layer has complete linearity and feeds into a softmax layer for classification. Data distribution is changed by each convolution layer during network training. The vanishing gradient problem occurs when the activation function for the neural net becomes saturated, and the parameters are no longer updated. This problem is typically solved by using rectified linear units (ReLUs) for activation. Batch normalization is a technique for increasing speed, performance, and stability. Each convolution result is treated through the batch normalization algorithm and the activation function ReLU. Stride convolution in depthwise convolutions handle downsampling for the first layer and the rest of the network. Final average pooling reduces spatial resolution to 1 and counts depthwise and pointwise convolutions as separate layers before fully connecting the layer. Furthermore, the deep and separable convolutional structure enables MobileNet to accelerate training and considerably reduces the number of calculations. The reasons are as follows:

First, the standard convolution structure can be expressed as follows: $D_F$ denotes the input feature picture side length, $D_K$ the convolution kernel side length, $M$ the number of input channels, $N$ the number of output channels, $K$ the convolution kernel, and $F$ the feature map. The output feature map for standard convolution assumes that stride one and padding are computed as follows:

$$G_{k,l,n} = \sum_{i,j,k} K_{k+i=1,l+j-1,m} \tag{2}$$

The computing cost is $D_K$, $D_K$, $M$, $N$, $D_F$, and $M$ and $N$ are the numbers of input and output channels, respectively. During standard convolution, the input image, including the feature image FM,

includes feature maps, which use the zero-padding fill style. Equation (4) is the depthwise separable convolutions cost. The first item is the depthwise separable convolutions calculation and the latter is the calculation of the pointwise convolutions.

$$D_K \cdot D_K \cdot M \cdot N \cdot D_F \cdot D_F + M \cdot N \cdot D_F \cdot D_F \tag{3}$$

$$\frac{D_K \cdot D_K \cdot M \cdot N \cdot D_F \cdot D_F + M \cdot N \cdot D_F \cdot D_F}{D_K \cdot D_K \cdot M \cdot N \cdot D_F \cdot D_F} = \frac{1}{N} + \frac{1}{D_K^2} \tag{4}$$

The deep separable convolution structure of MobileNet can obtain the same outputs as those of standard convolution with the same inputs. The depthwise phase requires $M$ filters with one channel and a size of $D_K \cdot D_K$. The Pw phase requires $N$ filters with $M$ channels and a size of $1 \times 1$. In this case, the computing cost of the deep separable convolution structure is $D_K \cdot D_K \cdot M \cdot N \cdot D_F \cdot D_F + M \cdot N \cdot D_F \cdot D_F$, approximately $\frac{1}{N} + \frac{1}{D_K^2}$ of that of standard convolution.

In MobileNet v1, the network layer design is fairly simple, and changes the convolution algorithm to reduce parameters. Compared with MobileNet v1, MobileNet v2 changes much of the network architecture and increases the residual layer structures. This layer is based on the convolution layer structures of the ResNet architecture, as in residual learning that employs shortcuts. These improve the accuracy of the depthwise convolution layer without having large overhead. Bottleneck layers that reduce input size are also used. An upper bound is applied to the ReLU layer, which limits the overall complexity. The structure of the standard convolution layers used in MobileNet v2 is illustrated in Figure 9.

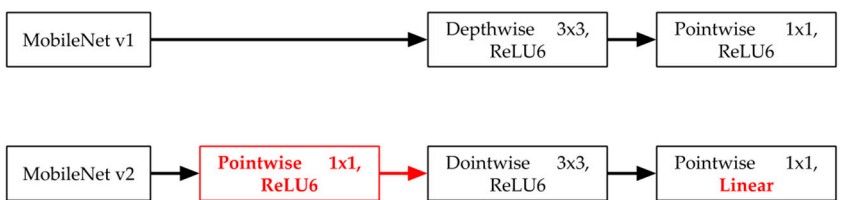

**Figure 9.** Comparison of convolutional blocks among architectures.

### 2.2.3. Vehicle Detection Based on MobileNet—YOLO Model

In this study, we implemented a YOLO v3 detection model with MobileNet v2, and the model considerably reduced parameters and gained high accuracy within limited hardware conditions. The initial weight of MobileNet v2 was trained in ImageNet; from this, we could obtain a great effect and use fewer samples to train YOLO. MobileNet is a base network for extracting image features, and the characteristics extracted using MobileNet v2 and upsampling contain three scales of object detection and modify the filter numbers from 512, 256, and 128 to 160, 96, and 64, which are the settings for MobileNet v2. All the calculations of the convolution layer are changed to depthwise and pointwise convolution. Compared with the original YOLO-based on Darknet53, the version implemented with MobileNet reduces the number of required parameters and improves the prediction speed.

MobileNet is only a feature extraction layer. In the implementation process, only the convolution layer in front of the pooling layer is used to extract features; therefore, multidimensional feature fusion and prediction branches are not reflected in the network.

### 2.3. IOU and Overlapping

We used the modified detection model to verify the accuracy of the CNRPark + EXT dataset. The parking grid provided in the dataset is presented in Figure 10. The dataset presented in Acharya and Khoshelham's approach for real-time car parking occupancy detection used a CNN to classify parking space status. The CNN model judged whether cars were in parking spaces after assigning bounding boxes to each parking space. However, a real parking grid was used to judge this reasonable;

therefore, we applied our method to automatically generate the actual parking gridline, as revealed in Figure 10b. The parking grid automatic generation method was provided in True's study [47], which provided a canny edge and Hough line to detect edges on real parking grids.

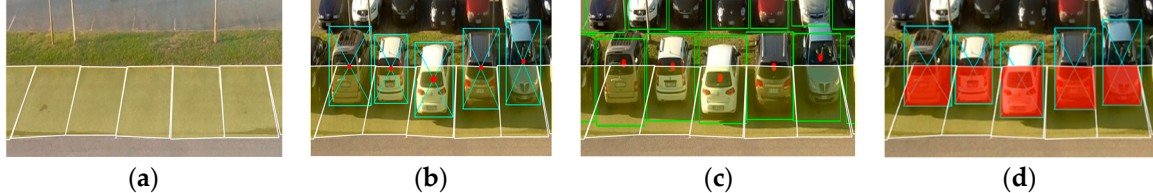

(**a**)　　　　　　　　(**b**)　　　　　　　　(**c**)　　　　　　　　(**d**)

**Figure 10.** Comparison of methods using center points and area. (**a**) Parking lots. (**b**) Green bounding boxes provided by the dataset determine parking-space occupancy. (**c**) Blue bounding boxes are detected using YOLO. (**d**) The overlapping and IOU method is used to determine parking occupancy.

In the parking dataset, we used the center points of bounding boxes, which were predicted using YOLO to create a comparison table by using the methods of Acharya and Khoshelham [20] and Ciampi et al. [25]. The model achieved considerable accuracy, but the center point may have misjudged actual parking spaces because of the network surveillance camera's perspective, as shown in Figure 10c. To resolve the problem of misaligned perspectives, we propose a method of YOLO detection and object overlapping identification to improve occupancy detection results and to identify the occupancy status with the voting mechanism. Our judgment method of the overlapping is presented below (see Figure 11).

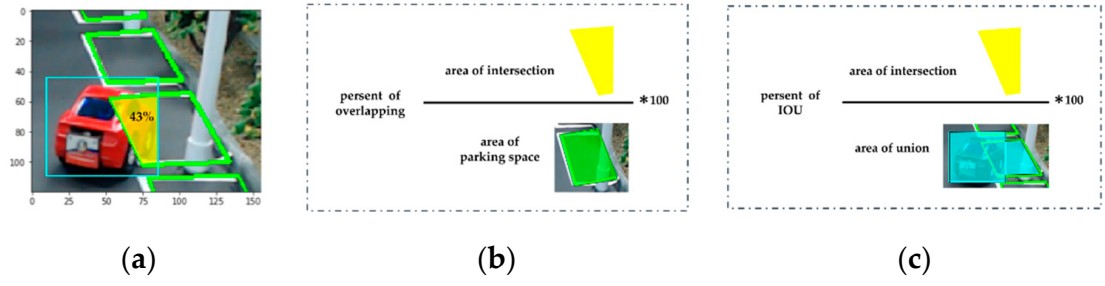

(**a**)　　　　　　　　　　(**b**)　　　　　　　　　　(**c**)

**Figure 11.** (**a**) Real-time situation of the overlapping area; (**b**) overlapping; and (**c**) IOU.

This step used YOLO to help find objects from the image, and could quickly determine object locations from the model. The camera angle caused difficulties in understanding if the item had passed the streetlight or parking space, and even determining which objects entered which parking grid was difficult. We used bounding boxes to calculate areas from the YOLO detection model, the area of each parking grid, the overlapping area ratio, and IOU. $B$ and $G$ represent two areas of object and parking grids, and $t$ represents threshold. $IOU(B, G)$ and $Overlapping(B, G)$ are defined as follows:

$$IOU\ ratio(B,\ G) = B \cap G \div B \cup G \tag{5}$$

$$Overlapping\ ratio(B,\ G) = B \cap G \div G \tag{6}$$

where $B \cap G$ is intersection area and $B \cup G$ the bounding box and parking space area. IOU calculation revealed parking grids closest to bounding boxes, and the overlapping ratio revealed the location of the largest ratio of bounding boxes. Furthermore, the IOU prevented cars in the front row from occluding with cars in the back. In other words, the IOU could compensate for overlapping limitations. When $IOU\ ratio(B,\ G) > t$, the car then found the closest parking grid. When $Overlapping\ ratio(B,\ G) > t$, the model found the parking space containing the bounding box.

IOU and overlapping determined the intersection area with percentages that inferred bounding boxes of cars occupying parking spaces. Therefore, we manipulated the camera angle to be set up when intersection area ratios were larger than the threshold of a certain area ratio. Therefore, we could determine objects entering grids. After cars were ensured to have stopped, the parking grids occupied by cars were judged. After obtaining each overlapping area ratio, we extracted the maximum value to enter the voting mechanism. In the voting mechanism, we used the voting concept to check whether a certain number of each five images showed a status that indicated that the space tended to be occupied. Figure 12 displays the schematic diagram.

## 2.4. Voting Mechanism

However, because of tilted camera angles, large vehicles passing through may have been counted as occupying the parking space with our method. Therefore, we proposed a voting method from time states for when vehicles and parking spaces reached the threshold value of the overlapping or IOU method, as presented in Figure 12. One point was used to denote the current frame of a parking space. If the points of the parking space reached a certain value within a certain time, we defined the parking space as occupied. Therefore, only when the parking space was occupied for a certain time would the parking space be defined as occupied. This could prevent the erroneous classification of parking spaces caused by large vehicles passing, and thereby, increase parking detection stability. Algorithm 1 is presented here to illustrate the logical architecture.

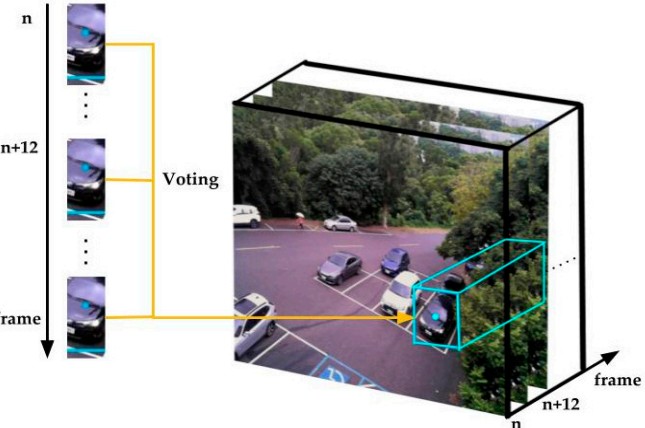

**Figure 12.** Illustration of voting mechanism.

## 2.5. Streetlight Judgment

During the day, we could record and process parking parts and not activate part of the streetlight until night, when the streetlight system could assist in parking and lighting. Settings were divided into two parts: (1) when no objects were detected on the street, the streetlight remained on a dim settings, and (2) when an object appeared or entered the streetlight's illumination range, the streetlight would brighten in advance of the object from the dim to the high stage, as shown in Figure 13. Smart streetlight automation may assist in parking safety and facilitate finding empty spaces.

---

**Algorithm 1.** Pseudocode for occupancy and voting mechanism.

---

1:  Input: $P\kappa \rightarrow$ bounding box of parking grid, $\forall \kappa = 1, \ldots, \kappa$
2:  Input: F $\rightarrow$ images of streaming camera
3:  Set number keep record the voting status, voting status
4:  Set number record voting status, V
5:  Set number of the threshold $\rightarrow$ T
6:  Set number Boolean $\rightarrow$ P
7:  While the streaming video does not stop:

   a.  bounding box F = YOLO.predict (F)
   b.  for bounding box F in every single $P\kappa$, $\forall \kappa = 1, \ldots, \kappa$

     i.  Intersection area $\rightarrow$ calculate Intersection area (bounding box F,$P\kappa$)
     ii.  Union area $\rightarrow$ calculate Union area (bounding box F,$P\kappa$)
     iii.  IOU ratio $\rightarrow$ Intersection area/Union area
     iv.  Overlapping ratio $\rightarrow$ Intersection area/area of $P\kappa$

   a.  end for
   b.  for every single $P\kappa$ , $\forall \kappa = 1, \ldots, \kappa$ finds which value is bigger than threshold by IOU or overlap:

     i.  if the value of IOU ratio bigger than the threshold T

      1. afterward, enter the voting mechanism, and P = 1
     ii.  else check the value of Overlapping ratio bigger than threshold T

      1. afterward, enter the voting mechanism
   c.  end for
   d.  enter the Voting mechanism:

     i.  if P == 1:

      1. update V from IOU
     ii.  else:

      1. update V from Overlapping
     iii.  end if
   e.  end the Voting mechanism

8:  per five F check the result of the Voting mechanism

   a.  if V keeps the certain status

     i. $P\kappa$ has been occupancy
   b.  end if

9:  end while

---

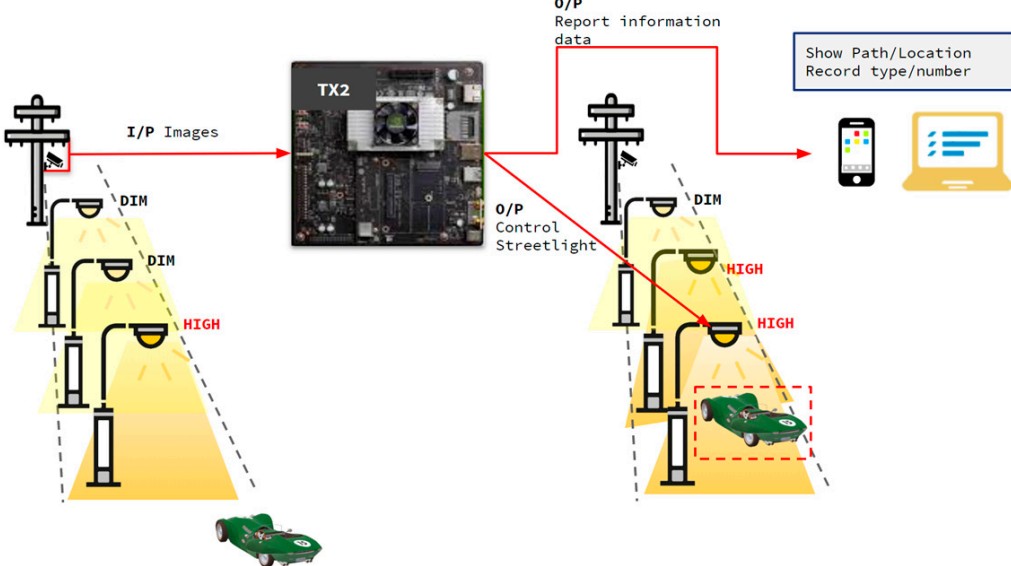

**Figure 13.** Illustration of smart streetlight system.

## 3. Experimental Results

In this section, the proposed method is verified separately in the model, dataset, and real scenes. To verify the performance of the proposed occupancy detection part and the smart streetlight control part, in Section 3.1 we evaluate a hand-made model which verified the smart streetlight control effect and the judgment accuracy. In Section 3.2, we compare our network with others using the same dataset. And lastly, in Section 3.3, we verify our proposed method in real situations to ensure the quality of its performance.

### 3.1. Model for Reality Situation

We created a model to simulate real scenes according to camera images detecting moving objects. The streetlight switched from the dim to the high state when a car entered a roadside parking space; then, the occupancy detection model detected and recorded when the car occupied the parking space. In the model, three LEDs on both sides of the street represent the streetlight. A 3.1–3.4 V white LED was soldered to a GPIO on the Jetson TX2. The status of parking grids was identified through the YOLO model, IOU, and overlapping judgment; the voting mechanism ensured that the target stopped and occupied the parking space. All these processes were executed on the Jetson TX2. Figure 14 depicts the final demonstration of the proposed model comprising two parts, namely the streetlight and parking occupancy detection sections. The automatic streetlight system switches to the dim setting at nighttime and the high setting during vehicle movement. In the daytime, no LEDs glowed. Time was the key that prompted the streetlight to switch between day and night settings. Figure 14a shows that at nighttime, no motion was detected by the camera, and therefore, the streetlight remained in the dim state. The proposed model, illustrated in Figure 14b–d, was designed only for the detection model to detect the object's presence from real-time images and glow more brightly; then, the remaining LEDs maintain their dim state. In Figure 14b, for instance, the first set of LEDs glowed at the high setting when an object was detected by YOLO, the second set glowed before the object approached, and the remaining LEDs stayed in the dim state because it was nighttime. Moreover, when the object moved to the second set, the second set of LEDs remained in the high setting, the first set reverted to the dim state, and the third set switched to the high state for prelight mode, as displayed in Figure 14c. These results demonstrate the efficiency of the proposed idea and validate the proposed model.

Figure 15 presents parking status detection using the identify model. In the sequence of images received from the camera, for each frame, the detection model detected objects and calculated the

IOU and overlapping ratio to each parking space area. In the sequence of images, the green grids represented free parking spaces and the red ones represented occupied spaces. Figure 15a reveals that the red car drove to one of the parking spaces, the detection model detected the red car intersecting with the parking space. Then, the red car received a point from the voting mechanism until the score of the red car exceeded the threshold. The detection model then switched the parking grid from green to red, and the car was considered to be occupying the parking space.

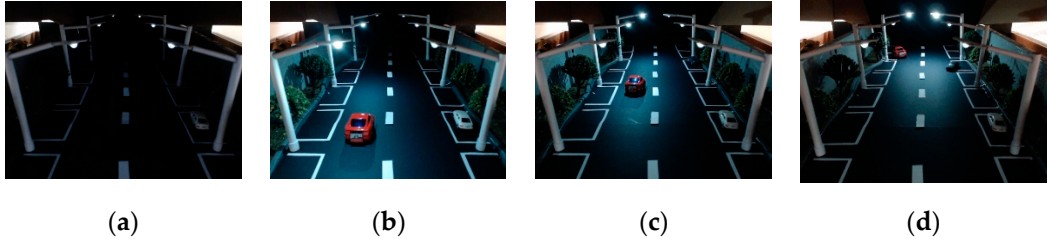

(**a**)  (**b**)  (**c**)  (**d**)

**Figure 14.** Images of the automatic streetlight control system, which switched from dim night settings to a high state during object detection. In the daytime simulation, the LEDs were not illuminated. (**a**) In the nighttime representation, the dim LEDs were illuminated. (**b**) When an object was detected by the detection model form camera, the first set of high LEDs was illuminated, whereas the rest remained in the dim mode. (**c**) Only the second set of LEDs glowed at the high setting, and (**d**) the rest remained in a dim state.

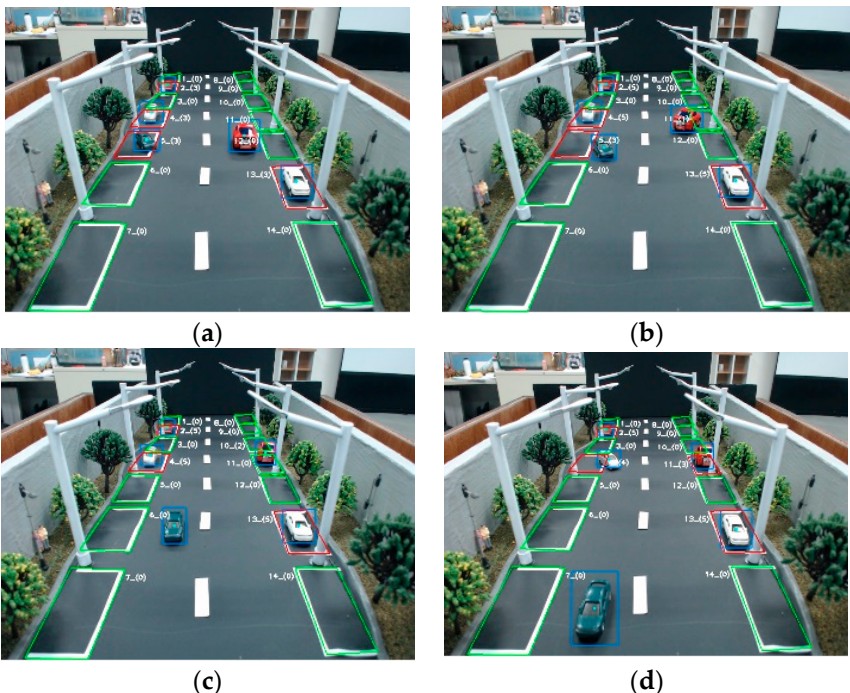

(**a**)  (**b**)

(**c**)  (**d**)

**Figure 15.** Streetlight control with the Jetson TX2. A sequence of images showed (**a**) the red car driving into the parking grid. (**b**) The model began to determine whether the red car occupied the space. (**c**,**d**) The voting mechanism verified that the red and blue cars stopped at parking grids.

*3.2. Network*

To illustrate the effectiveness of the proposed algorithm, from the CNRPark + EXT dataset divided into two sets, we randomly selected approximately 205 images as our training set, which contained three dates and weather conditions for nine cameras. For the testing set, the image annotation tool LabelImg was used to label six types of classes, and the obtained images were used as the ground truth for testing.

In this study, we compared our method with other current object detection algorithms including Mask R-CNN, sliding window, and CNN. Tables 1 and 2 present the accuracy and the mean occupancy error (MOE) [25] among the networks. The sliding window and CNN used the window sliding method to scan whole images to obtain parking space occupancy status. Although this method achieved high accuracy on the dataset, detection with the sliding window was slow. Mask R-CNN adopted the risk priority number method to obtain the candidate bounding boxes, and both accuracy and detection speed improved significantly. However, the requirements for real-time detection were still not met. Object detection and localization through YOLO were performed within one step, and high-accuracy real-time processing was achieved.

**Table 1.** Detection accuracy of our YOLO + IOU + Overlapping method on CNRPark + EXT.

| Name of Method | Testing Accuracy |
|---|---|
| AlexNet | 96.54% |
| ResNet50 | 96.24% |
| CarNet [21] | 97.24% |
| Ours | 98.97% |

**Table 2.** Detection MOE of our method.

| Name of Method | MOE |
|---|---|
| mAlexNet | 4.17% |
| Mask R-CNN | 5.23% |
| Re-trained Mask R-CNN [25] | 3.64% |
| Ours | 1.56% |

For testing purposes, the test subset of the CNRPark + EXT dataset was used. The accuracy is used to evaluate the performance of the identification task. It is calculated as the number of all correct predictions divided by the total number of the dataset and the best accuracy is 1.0, which is calculated as follows:

$$\text{Accuracy} = \frac{T_P + T_N}{T_P + T_N + F_N + F_P} \tag{7}$$

where $T_P$, $F_N$, $F_P$, and $T_N$ represent the number of true positives, false negatives, false positives, and true negatives, respectively.

Following other counting benchmarks, we used MOE, where $N$ is the total number of test images, $C^{gt}$ is the actual count, $C^{pred}$ is the predicted count of the $n$-th image, and $num\_slots_n$ is the total number of parking lots in the current scene. This evaluation metric is expressed as a percentage, defined as follows:

$$\text{MOE} = \frac{1}{N} \sum_{n=1}^{N} \frac{\left| C_n^{gt} - C_n^{pred} \right|}{num\_slots_n} \tag{8}$$

The purpose of our method was to identify occupancy status on parking grids, and therefore, high accuracy of bounding boxes fitting objects is crucial for calculations of the area.

The precision and recall value of YOLO v3 on the CNRPark + EXT dataset was 99.81% and 98.64%, respectively, and the average processing time of each image was 41 ms on the Jetson TX2. For the Jetson TX2, we used MobileNet v2 to reduce parameters and release CPU memory. Table 3 depicts the compared frames per second (FPS). The number of parameters in Darknet-53 was 14 times that of MobileNet v2. In theory, the parameters are simplified 14 times, and the speed significantly improves. However, Table 3 reveals that this only resulted in an upgrade from 2.3 to 1.8 FPS on the Jetson TX2, which only increased the speed by 1.3 times, because the implementation of the depthwise convolutional algorithm had not been well optimized on MobileNet.

**Table 3.** MobileNet v2 compared to Darknet-53 by per frame, FPS, and parameter.

| Name of Network | Per Frame | FPS | Parameter |
|---|---|---|---|
| Darknet-53 | 0.55 sec | 1.8 | 61576342 |
| MobileNet v2 | 0.41 sec | 2.3 | 4359264 |

Details of the hardware are briefly reported for the sake of completeness. The camera module was a Logitech-C920 HD PRO WEBCAM and Microsoft LifeCam Studio V2 (Q2F-00017) that supported 1080p30 video mode. The same method and different scenes at different resolutions are presented in Table 4. We compared our detection method with other scenes and resolutions including CNRPark + EXT, the real scenes 1 (daytime), and the real scenes 2 (nighttime). As shown in Table 4, the model maintains great performance under the slight brightness from the streetlight at nighttime. On most webcam images, the model can maintain approximately 90% accuracy in any weather conditions.

**Table 4.** Model performance in different scenes and resolutions.

| Image | Camera Model | Image Size (H) × (V) | Resolution | Accuracy |
|---|---|---|---|---|
|  CNRPark + EXT | 5 MP Fixed Focus Color Camera | 2592 × 1944 pixels | 96 dpi | 98.97% |
|  real scenes 1 (daytime) | Logitech-C920 HD PRO WEBCAM | 1204 × 907 pixels | 142 dpi | 96.71% |
|  real scenes 2 (nighttime) | Microsoft LifeCam Studio V2 (Q2F-00017) | 1920 × 1080 pixels | 96 dpi | 89.97% |

### 3.3. Verifying Our Approach with Real Scenes

The parking grid judgment process is presented in Figure 16. First, the original image (see Figure 16a) was fed as an input to the model. After finding the parking spaces (see Figure 16b), the detection model drew center points and bounding boxes (see Figure 16c). Judgment of occupancy used the IOU area and overlapping methods (see Figure 16d). Judgment of parking space occupancy was based on a series of images. The cumulative method was used to calculate which parking space a car belonged to, and each parking space had its own voting accumulator. Which car belonged to which parking space was determined through the voting mechanism, which could filter noise when cars crossed into and out of parking spaces.

The experiments took different environments into account, i.e., cases 1 to 5. Cases 1 to 3 are daytime settings. There is a car parking/ready to leave/pass through other parking spaces in different situations of occupancy detection. Cases 4 to 5 are nighttime settings; however, there is a streetlight on one side (parking space 6–9) and none on the other. The environment is side facing, complex, and full of obstructions.

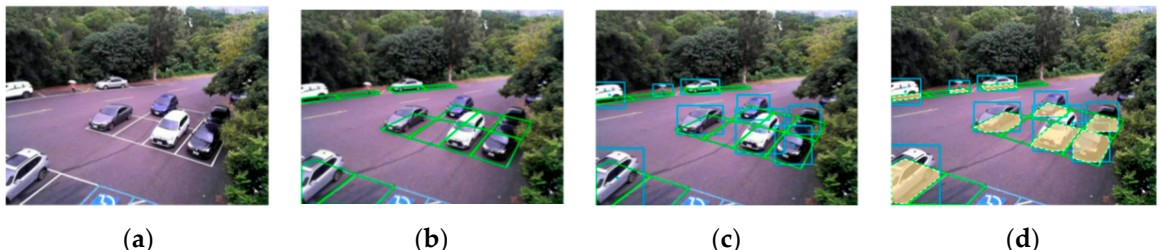

**Figure 16.** Occupancy detection process. (**a**) Original image of the input. (**b**) Green bounding boxes represent parking grids. (**c**) Blue bounding boxes from YOLO represent detected vehicles. (**d**) The yellow area was the intersection between parking grids and vehicles.

In case 1, Figure 17(a1–a4) demonstrates that a car approached parking space 1. Parking spaces in red represent occupied spaces, those in green represent free spaces, and those in blue represent bounding boxes of the predict model and signified vehicles intersecting with parking spaces. The model continued to detect and compare intersection areas to verify that vehicles were parked in parking spaces. Therefore, this model continued judgment through the voting mechanism over time. Case 2 is presented in Figure 17(b1–b4). The car in parking space No. 10 was ready to leave. The model identified that parking space 10 was occupied and continued to detect and compare the intersection area to ensure that the vehicle remained in the parking space. The model detected that the intersection area with the parking space was becoming smaller until the car completely left, after which the model detected that the parking space was empty and switched parking space 10 from red to green, thereby identifying the space as free. Case 3 is presented in Figure 17(c1–c4). It verified through the voting mechanism that cars did not affect judgment when passing through the parking space.

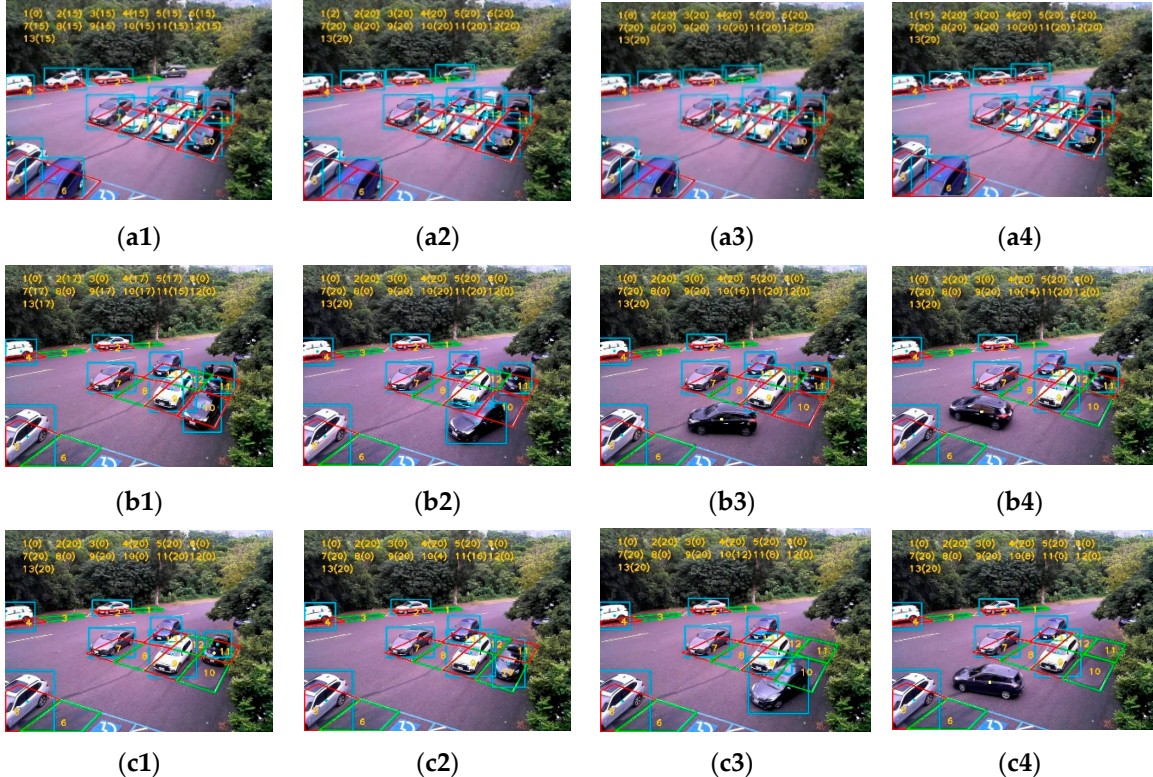

**Figure 17.** Several conditions with the proposed method. (**a1**–**a4**) A car parking. (**b1**–**b4**) A car leaving its parking space. (**c1**–**c4**) A car passing through other parking spaces while leaving its parking space.

Case 4 is presented in Figure 18(a1–a6). The results show that the proposed system can effectively detect the occupancy of parking spaces at nighttime. In case 5, the detection model might cause misjudgments when vehicles are blocked by trees, as shown in Figure 18(b1–b6). In this study, the parking spaces can easily be detected by integrating the smart control system that can detect parking spaces automatically.

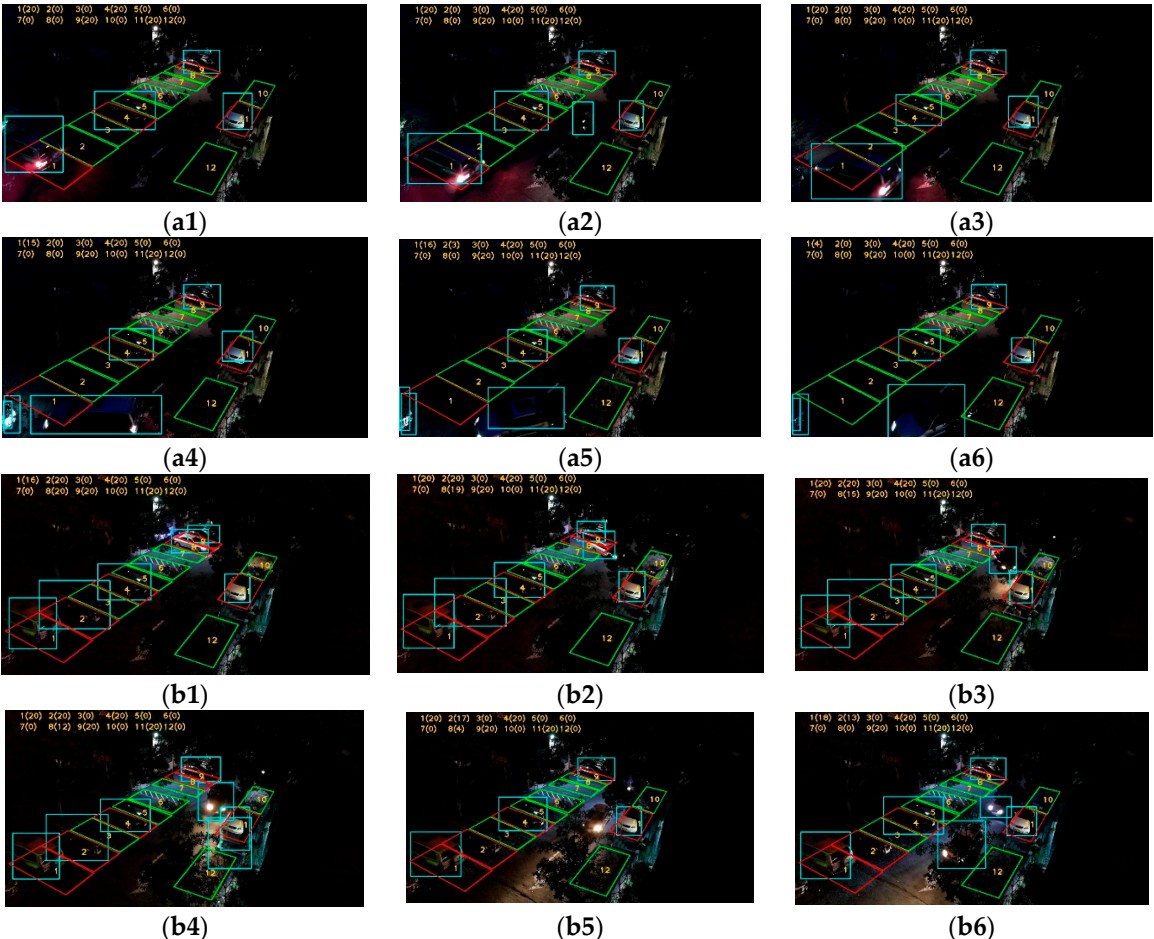

**Figure 18.** Nighttime conditions with the proposed method. (**a1**–**a6**) A car leaving parking space 1. (**b1**–**b6**) A car leaving parking space 8.

## 4. Discussion

Current deep learning solutions often require large amounts of computational resources to run. Running these models on embedded devices can lead to long runtimes and large resource consumption—including CPU, memory, and power—for even simple tasks [48]. By using Darknet-16 before adopting MobileNet, we reduced the number of network layers to 16 by reducing the residual block. Although many parameters were reduced, speed increased, and little precision was lost with Darknet-16. This lightweight network still had 20,000,000 parameters. On the Jetson TX2, the energy consumption was proportional to the model inference time. The model sped up overall inference and reduced energy consumption by nearly 1.5 times compared with MobileNet and Darknet-16. The number of parameters with our model of MobileNet was small, being six times lower than that of Darknet-16. However, speed did not increase proportionally to the number of parameters because depthwise convolution was realized through group convolution. Therefore, the speed and number of parameters was not proportional, and reducing computational cost was not necessarily the same as increasing speed, which depends on the implementation and how it can be parallelized.

Although our system worked effectively the first time, this occupancy identification method only used overlapping to determine the ratio area through a sequence of images. Interobject occlusion made the voting yield erroneous results. Figure 19 provides an overview of the control group. In Figure 19a, an input image appeared of a white car in front of a navy blue one. We calculated the overlapping area of the navy-blue car to be larger than half, as shown in Figure 19b. The overlapping area of the white car nearly covered the entire parking space in Figure 19c. Figure 19b,c reveals that IOU usage verified the parking grids containing cars. The IOU allowed the model to recognize cars close to parking grids, although the overlapping areas in these parking grids were all larger than half. In these cases, the overlapping ratio area caused erroneous identification. Thus, IOU could offset the overlapping. In IOU, the navy-blue car was closer to the parking space because the union area of the white care was larger than that of the navy-blue car.

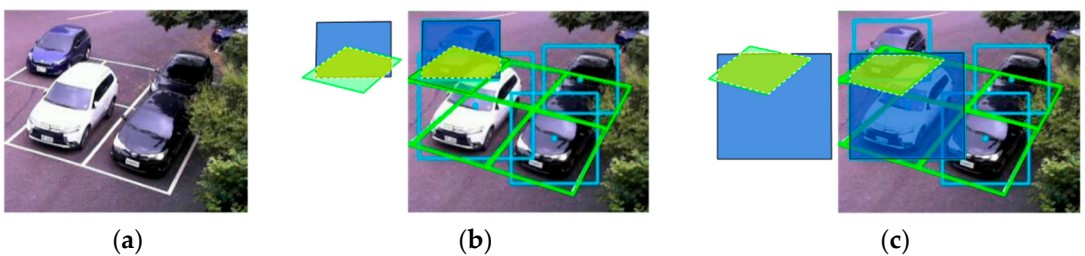

(**a**)  (**b**)  (**c**)

**Figure 19.** Overlapping and IOU comparison. (**a**) Original image. (**b**) Overlapping area of the car in the back and the parking space. (**c**) Overlapping area of the car in the front and the parking space.

## 5. Conclusions

In this paper, we proposed a lightweight parking occupancy detection system. A large number of sensors could be replaced by only one embedded device. The cost of the device is low, and it can be quickly deployed on the streets. Our system detects vacant parking spots and controls streetlights, which decreases the amount of time people need to find parking spaces on streets. Our system also changes streetlight brightness according to various circumstances to reduce the potential waste of resources at night.

For the real scene's applications, we prepared two different scenarios that could manage off- and on-street parking, with a smart control system. Tests in the real environment showed that the system works well under a wide range of light conditions. Otherwise, during the night, streetlights support the detection system, providing on-demand adaptive lighting, and making streetlights adjust their brightness based on the presence of pedestrians or cars to stay safe. This system can contribute to the creation of smart streets. The use of our intelligent system could help governments to collect potential data and to turn it into big data for future analyses of road conditions and even land management.

This proposed parking occupancy detection system is based on object detection. We designed an object detection model using the YOLO v3 algorithm derived from MobileNet v2. This model is lightweight and can operate on the embedded system Jetson TX2. The embedded system can easily be built into existing streetlights, and one device serving many parking grids can decrease maintenance requirements. IOU and overlapping were proposed to determine parking space occupation, replacing the typical classification method. For parking occupancy detection, accuracy and recall using the CNRPark + EXT dataset reached 99% and 95%, respectively, indicating that the model can completely detect parking-space occupancy. Voting was applied to verify this system, allowing the system to detect occupancy under any condition.

Regarding the control of streetlight brightness, object detection was used to detect vehicles on the street at night; accordingly, streetlights would enter dim mode when no vehicles were detected on the roads, thus saving energy. Experiments demonstrated that our parking occupancy detection system performs well under various weather conditions in both daytime and nighttime.

**Author Contributions:** All authors were involved in the study presented in this manuscript. C.-H.T., L.-C.C. and R.-K.S. determined the research framework, proposed the main idea, performed mathematical modeling, and contributed to revising and proofreading the article. W.-Y.P. performed algorithm implementation and simulation and wrote the original manuscript. J.-H.W. reviewed the manuscript and provided scientific advice. All authors have read and agreed to the published version of the manuscript.

**Funding:** This research received no external funding.

**Conflicts of Interest:** The authors declare no conflict of interest.

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
