# Peer review of "Video-Based Parking Occupancy Detection for Smart Control System"

_applsci, doi:10.3390/app10031079_

Round 1

Reviewer 1 Report

the paper "video-based parking occupancy detection for smart control system" by Chen et al. proposed an efficient parking management system. They authors have proposed novel technique to use one embedded device in place of large number of sensors. Their proposed sensors are claimed to be cheap and easy to install. Using the system one can identify available parking areas, manage lights on streets according to the requirement or control the light brightness using the modern algorithms of deep-learning. Overall the proposed method is efficient and holds promise to create smart streets in modern cities. The presentation and the scientific research work are carried out well. Based on these facts I advocate to let this manuscript to be published in its present form.

Author Response

Thanks for the your recommendation. We have reviewed all the content, and added nighttime conditions for experiment results. The experimental results are shown in Section 3.3. Please refer to the revised version.

Reviewer 2 Report

This paper presents an approach for the detection of lightweight parking occupancy. This work is well written but shows the following weaknesses;

Please, add information about the valuation of experts in this field: useful, time of answer, etc. A comparison vs. actual approaches, although with different technology can be added. Please, show more results for different environments under different conditions. Which is its robustness at night?? Which is the effect for the use of different cameras?

This work is very interesting, but it can be improved.

Author Response

Many thanks for the suggestion. We have added more content and experiment to present our proposed system in the revised version. Please kindly find the attached file.

Reviewer 3 Report

This is a very interesting paper that is well presented and well written. I only have one substantive comment about the content. I think the conclusions can use a bit more discussion about related applications or uses of the methods. How does the approach generalize to associated topics like curb management, security, traffic optimization, etc.

Author Response

Thanks for the suggestion. The similar studies and related applications have been added in Introdection and Conclusion section. Please refer to the revised version.

Round 2

Reviewer 2 Report

This paper presents an approach for the detection of lightweight parking occupancy. This work has been improvement but I can improve a bit more. In particular, authors can add more comments about the use of different resolution due to use of different cameras. Or at least, to show the robustness of the proposal in order to be applied with different resolutions.

The rest of the comments have addressed well.

Author Response

Many thanks for the suggestion. We have added the comparison results to present the proposed method in different resolution and scenes, as shown in Table 4. The results are described in Line 489-495. Please refer to the revised version.
